# CAUSAL DISCOVERY WITH UNOBSERVED VARIABLES: A PROXY VARIABLE APPROACH

## ABSTRACT

Discovering causal relations from observational data is important. The existence of unobserved variables, such as latent confounders or mediators, can mislead the causal identification. To address this issue, proximal causal discovery methods were proposed to adjust for the bias with the proxy of the unobserved variable. However, these methods only focused on *discrete* variables, which limits their real-world application. Besides, the extension to the continuous case is not easy as the naive discretization method can introduce biases due to the discretization error. To tackle this challenge, we propose a new method based on a comprehensive analysis regarding discretization error. We begin by identifying the source of discretization error and how it introduces the bias. We then introduce smoothness conditions under which the discretization error can be reduced to an infinitesimal level, provided the proxy is discretized with sufficiently fine bins. We also find that such conditions can hold for a broad family of causal models, *e.g.*, Additive Noise Model. Based on this, we design a proxy-based hypothesis test that is provable to be consistent for identifying causal relationships within continuous variables. We demonstrate the utility of our method on synthetic and real-world data.

## 1 INTRODUCTION

Causal discovery has received increasing attention due to its wide application in machine learning, medicine, and psychology. While the golden standard for causal discovery remains randomized experimentation (RCT), it can be too expensive, unethical, or even infeasible to conduct in the real world. For this reason, identifying causal relations from pure observation data is desirable.

A well-known issue of inferring causation from observation is the existence of unobserved variables, such as latent confounders Jager et al. (2008) and mediators Hicks & Tingley (2011). These variables describe intrinsic characteristics that are widely present but difficult to quantify (*e.g.* health status, living habit), and hence induce biases for the causal identification.

To address this problem, recent studies proposed to leverage the proxy variable to correct for the bias caused by the unobserved variable. Here, the proxy variable can be a noisy measurement or an observed child of the unobserved variable, from which useful information about the latent bias can be inferred. For example, (Rothman et al., 2008) developed a matrix adjustment method based on prior knowledge of the proxy generation mechanism; (Kuroki & Pearl, 2014) showed that the generation mechanism could be automatically identified, thus avoiding the need for external information. Based on this, (Miao et al., 2018; Tchetgen et al., 2020) proposed a proxy-based hypothesis test to distinguish causation from latent confounding, and Tchetgen Tchetgen et al. (2023) treated the proxy as a negative control outcome and employed control outcome calibration Tchetgen Tchetgen (2014) for identification.

Despite the progress that has been made, these works only focused on discrete variables, which constrained their applicability in scenarios involving continuous variables, such as medication dosages and vital sign measurements. A natural approach for such scenarios is discretizing continuous variables into discrete ones and applying the proximal method for discrete variables. However, this approach can result in misidentification Liu et al. (2002); Margaritis (2005), because the discretization errors can break the independence structure brought by the proxy variable, *i.e.*, a key property for the proximal method to work, as illustrated by the following example. Therefore, it remains an open question of *how to infer causal relations for continuous variables*.

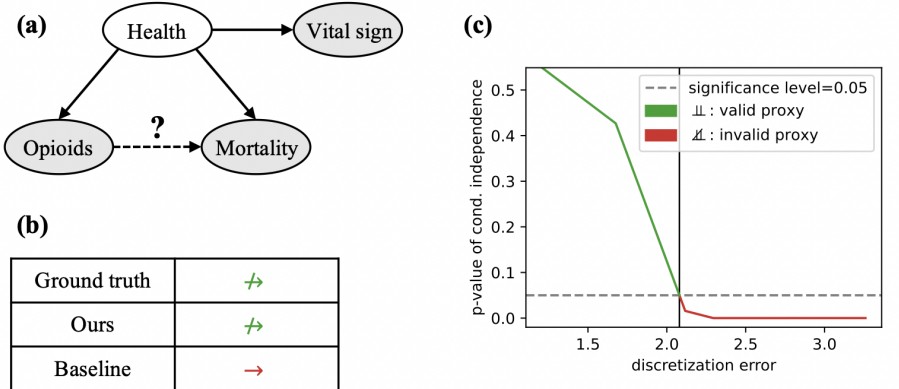

Figure 1: Illustration of Exam. 1.1. (a) The causal graph where observed variables are marked in gray. (b) Identification results of our method and Miao et al. (2018) for discrete variables. (c) As the discretization error grows, the identification changes from valid (green) to invalid (red).

**Example** 1.1. Prescription opioids (*e.g.*, morphine, oxycodone) have been traditionally used as pain relievers in patients with severe disease McQuay (1999). Despite their wide usage, critics often question their safety, citing a higher mortality rate associated with opioid use Zhang et al. (2018). Indeed, this association can be a confounding bias induced by the hidden health status, according to recent RCTs Anand et al. (2004); Rosini et al. (2015), in the sense that patients with poorer health often require more opioids to ease the pain, and meanwhile are more likely to die. To remove the bias, we can use the patient's vital sign, *e.g.*, blood pressure, as the proxy variable. However, as shown in Fig. 1 (b), naively applying the discrete method ("Baseline") leaves the bias remaining. This is because the discretization error can break key independence in the proximal model (Fig. 1 (c)) and therefore compromise the power of the proxy variable.

In this paper, we propose to tackle this challenge with a proximal-based method grounded in a comprehensive analysis of discretization methods. Specifically, we first propose to identify the source of the discretization error as well as how it introduces the identification bias. We further identify smoothness conditions under which this error can be reduced to arbitrarily small. Moreover, we pleasingly find that this condition can hold for a broad family of causal models (*e.g.*, Additive Noise Model), as long as the structural equation is continuous and differentiable. Based on this analysis, we proceed to infer the causal relations with a hypothesis testing procedure that is provable to be consistent, given a sufficiently large sample size and fine and finite partitions of the continuous variables. Back to the opioids example, Fig. 1 (b) shows that our identification result aligns well with the RCT findings, while compared baselines fail to do so.

Our contributions are summarized as follows:

1. We identify smoothness conditions to effectively control the discretization error.

2. We propose a proximal-based hypothesis testing method that is consistent for causal identification.

3. We demonstrate the validity and utility of our method on synthetic and real-world data.

## 2   RELATED WORKS

**Proximal causal discovery**. The use of proxy to adjust for latent bias can be traced back to the seminal work of (Rothman et al., 2008), who developed a matrix adjustment method that required external knowledge about the proxy generation mechanism. To ease this requirement, (Kuroki & Pearl, 2014) showed that for surrogate-rich setting, *i.e.*, cases with multiple proxies, one could identify the error mechanism and thus the causal relation without external information. Following this work, the surrogate-rich setting has been extensively studied in (Louizos et al., 2017; Deaner, 2021; Mastouri et al., 2021; Miao et al., 2022; Cheng et al., 2022). On the other side, identification with a single proxy, though of higher practical value, has been hard to come by. For example, the method

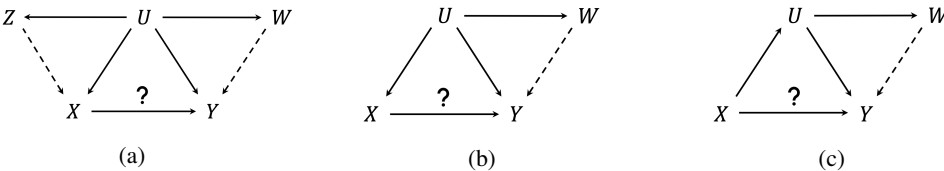

(a)                                    (b)                                    (c)

Figure 2: Illustration of the proximal causal discovery. $X \overset{?}{\to} Y$ is the causal edge of interest. $U$ is the unobserved variable. $W$ and $Z$ are *resp.* the outcome-inducing and the treatment-inducing proxy variables, where $W \dashrightarrow Y$ and $Z \dashrightarrow X$ mean the edges can exist or not. (a) denotes the graph for discrete variables in Miao et al. (2018). For continuous variables, our method only requires a single proxy to test the causal relation with confounding bias in (b) and mediation bias in (c).

in (Tchetgen Tchetgen et al., 2023) could not achieve complete identification. (Miao et al., 2018; Tchetgen et al., 2020), the works most related to ours, only applied to discrete variables satisfying particular level constraints. **In our paper**, we propose a novel approach that can achieve complete identification for the continuous case with a single proxy.

**Conditional Independence (CI) test with discretization.** Causal discovery is closely related to the test of CI, since the correlation between two variables is believed to be causation only if it does not disappear conditioning on a third variable Pearl (2009). Since testing CI is much easier in the discrete case, many attempts have been made to test continuous CI via discretization Margaritis (2005); Huang (2010). Nonetheless, general theoretical guarantees for such discretization were not carefully studied until the recent work of (Neykov et al., 2021). In (Neykov et al., 2021), the authors imposed smoothness assumptions on the conditional distributions, so that if one binned the conditioning variable with sufficient refined partitions, the problem of testing continuous CI could be converted (with tractable cost) to that of testing discrete CI in each partition. Such an idea is further exploited under weaker assumptions in (Warren, 2021). However, these methods required the conditioning variable to be observable. **As a contrast**, we study cases where the conditioning variable is unobservable and hence can not be directly discretized.

## 3 PRELIMINARY

**Problem setup.** We consider the problem of identifying the causal relationship between a pair of variables, $X$ and $Y$. The system also includes an unobserved variable $U$, which can be either a sufficient confounder or mediator between $X$ and $Y$. For this case, causal identification is generally not feasible due to the confounding Jager et al. (2008) or mediation bias Hicks & Tingley (2011). To adjust for the bias, we assume the availability of a proxy variable $W$. In practice, the proxy variable can be a noisy measurement or an observed child of the unobserved variable $U$. We illustrate the relations between $X, Y, U, W$ with models (b) and (c) in Fig. 2.

**Notations.** We use $\mathcal{P}_X, \mathcal{P}_{X,Y}, \mathcal{P}_{X|Y}$ to *resp.* denote the marginal, joint, and conditional distributions. We use $pr(x)$ to denote the density of $X$ (*resp.*, probability of $X = x$) for the case when $X$ is continuous (*resp.*, discrete). We then use $pr(X \in A)$ and $pr(Y \in B|x) := \int_B pr(y, x)/pr(x) dy$ to denote the probabilities of $X \in A$ and $Y \in B$ given $X = x$. For discrete $U, W$ with *resp.* $J, K$ levels, we use $P(W|u) := \{pr(w_1|u), ..., pr(w_K|u)\}^T$ and $P(w|U) := \{pr(w|u_1), ..., pr(w|u_J)\}$ to denote the column and row vectors consist of transition probabilities. The transition matrix from $U$ to $W$ is defined as $P(W|U) := \{P(W|u_1), ..., P(W|u_J)\}$. We use $\to_p$ (*resp.*, $\to_d$) to denote convergence in probability (*resp.*, in distribution). We denote $\|A - B\|_{L_1} := \sum_{i,j} |A(i,j) - B(i,j)|$ for any matrices $A, B$. We define $A \preceq$ (resp., $\succeq$)$B$ if $A(i,j) \leq$ (resp., $\geq$)$B(i,j)$ for any $i, j$.

**Hypothesis Testing in the discrete case.** When the system is composed of discrete variables, (Miao et al., 2018) proposed to identify the causal relation by testing the null causal hypothesis $\mathbb{H}_0 : X \perp\!\!\!\perp Y|U$, which will be rejected if the direct causal relation $X \to Y$ exists. To test $\mathbb{H}_0$ under the presence of unobserved $U$, (Miao et al., 2018) required another proxy variable $Z$ and proposed to test the linearity between two probability functions of $x$, *i.e.*, $pr(y|Z, x)$ and $P(W|Z, x)$, by exploiting the independence structure brought by $W$ and $Z$. Specifically, suppose that $X, U, W, Z$

are discrete variables with (*resp.*) $I, J, K, M$ levels such that $IM > K$, we have

$$P(W|Z, x) = P(W|U)P(U|Z, x), \ \ pr(y|Z, x) = P(y|U, x)P(U|Z, x),$$

since $W \perp\!\!\!\perp Z, X|U$. When $P(W|U)$ is invertible, we have the following:

$$pr(y|Z, x) = P(y|U)P(W|U)^{-1}P(W|Z, x) + \underbrace{\{P(y|U, x) - P(y|U)\}P(W|U)^{-1}P(W|Z, x)}_{= \ 0 \text{ when } \mathbb{H}_0 \text{ holds}}.$$
(1)

Under $\mathbb{H}_0$, we have $P(y|U, x) - P(y|U) = 0$ for any $x$. Therefore, the 2nd term in Eq. 1 disappears, leading to a linear relation between $pr(y|Z, x)$ and $P(W|Z, x)$. Based on this, (Miao et al., 2018) proposed to test $\mathbb{H}_0$ by measuring the linearity between $q_y := P(y|Z, X)$ and the covariates matrix $Q := P(W|Z, X)$. Specifically, with the estimated $\hat{q}_y$ and $\hat{Q}$,

$$\sqrt{n}(\hat{q}_y - q_y) \to_d N(0, \Sigma_y), \ \hat{Q} \to_p Q, \hat{\Sigma}_y \to_p \Sigma_y, \text{ with } \hat{\Sigma}_y, \Sigma_y \text{ positive-definite,} \qquad (2)$$

we can calculate $\xi_y := \{I_n - \hat{\Sigma}_y^{-\frac{1}{2}}\hat{Q}^T(\hat{Q}\hat{\Sigma}_y^{-1}\hat{Q}^T)^{-1}\hat{Q}\hat{\Sigma}_y^{-\frac{1}{2}}\}\hat{\Sigma}_y^{-\frac{1}{2}}\hat{q}_y$ as the least square residual of regressing $\hat{\Sigma}_y^{-\frac{1}{2}}\hat{q}_y$ on $\hat{\Sigma}_y^{-\frac{1}{2}}\hat{Q}$. If $\mathbb{H}_0$ holds, Miao et al. (2018) showed that $T_y \to \chi_r^2$, where $r := IM - K$. Then one can perform hypothesis testing using $T_y$ given any significance level $\alpha$.

**Challenges for the continuous case.** Extending this analysis to the continuous case is not easy, since Eq. 1 transforms into an integral form over the variable $U$, rendering linear regression and Eq. 2 infeasible. A natural approach is to discretize $X$ and $W$ and apply the above procedure. However, such a discretization, if not conducted properly, will break the independence structure $W \perp\!\!\!\perp X|U$ induced by the proxy variable $W$. This will introduce a non-ignorable error for hypothesis testing, as illustrated in Example 1.1 where $I = 3, K = 2, L := |Y| = 3$. Due to these challenges, *it remains open to identify causal relationships for continuous variables*.

**To tackle this issue**, we conduct a thorough analysis of the discretization error, which enables us to effectively infer the causal relationships involving continuous variables. In particular, we delve into how the discretization error matters and leverage Total-Variation (TV) Smoothness to demonstrate its asymptotic diminishment. With this guarantee in place, we proceed to show that the hypothesis test remains asymptotically valid. As a byproduct, our analysis only requires a single proxy $W$, instead of two proxies $W, Z$ in the discrete case. This practical implication is particularly valuable in situations where access to two proxies is not feasible.

## 4 METHODOLOGY

In this section, we introduce our method for testing the null causal hypothesis involving continuous variables when we can only access a single proxy $W$. Our method is based on a comprehensive analysis regarding the error introduced during discretizing the continuous domains of $X, W, Y$. In particular, we identify the smoothness conditions on the conditional distribution $\mathcal{P}_{Y|U,X}$ and $\mathcal{P}_{W|U,X}$, under which the discretization error can be controlled to an infinitesimal level, making the hypothesis testing remain valid even after discretization. We further show that such smoothness conditions can hold for a broad family of causal models, such as the Additive Noise Model (ANM).

The rest of the section is organized as follows. First, in Sec. 4.1, we delve into the discretization error to investigate where it occurs and how it matters. Then, in Sec. 4.2, we discuss how to control these errors with smoothness regularities. Finally, in Sec. 4.3, we show that the proper control of the discretization errors leads to an asymptotically valid hypothesis test.

### 4.1 IDENTIFYING SOURCES OF THE DISCRETIZATION ERROR

We provide an overview of our testing procedure and identify where discretization errors may occur and how they affect the procedure. Our procedure begins with binning the continuous $X, Y, W$ into their discrete copies $\tilde{X}, \tilde{Y}, \tilde{W}$, followed by the hypothesis testing on these copies.

To be specific, for continuous variables $X, U, W$ with bounded support, let $\{A_i\}_{i=1}^I, \{B_j\}_{j=1}^J$, $\{C_k\}_{k=1}^K$ be *resp.* the measurable partitions of $\mathrm{supp}(X), \mathrm{supp}(U), \mathrm{supp}(W)$, and denote the parti-

tioned discrete variables *resp.* as $\tilde{X}, \tilde{U}, \tilde{W}$. By the law of total probability, we have:

$$pr(W \in C_k|x) = \sum_j pr(W \in C_k|U \in B_j, x)\, pr(U \in B_j|x),$$

$$pr(y|x) = \sum_j pr(y|U \in B_j, x)\, pr(U \in B_j|x).$$

In the form of transition matrices, these mean:

$$P(\tilde{W}|x) = P(\tilde{W}|\tilde{U}, x)P(\tilde{U}|x), \ \ pr(y|x) = P(y|\tilde{U}, x)P(\tilde{U}|x).$$

Similar to the discrete case, we suppose that The matrix $P(\tilde{W}|\tilde{U}, x)$ is invertible for all $x \in \mathrm{supp}(X)$.

*Remark* 4.1. This condition requires the dependency of the proxy $W$ on $U$ to be strong enough, which is necessary for the proximal causal learning Kuroki & Pearl (2014); Miao et al. (2018).

Under Asm. 4.1, we can combine the two equations together by inversing the $P(\tilde{W}|\tilde{U}, x)$ and have:

$$pr(y|x) = P(y|\tilde{U}, x)\mathbb{P}(\tilde{W}|\tilde{U}, x)^{-1}P(\tilde{W}|x).$$

Heuristically, under $\mathbb{H}_0 : X \perp\!\!\!\perp Y|U$ and the independence structure $X \perp\!\!\!\perp W|U$ brought from the proxy $W$, we have $p(y|u, x) = p(y|x)$ and $p(w|u, x) = p(w|x)$ for any $x, u, y$. If we further have $P(y|\tilde{U}, x) \approx P(y|\tilde{U})$ and $P(\tilde{W}|\tilde{U}, x) \approx P(\tilde{W}|\tilde{U})$, then we can rewrite the above equation as[1]:

$$pr(y|x) \approx P(y|\tilde{U})P(\tilde{W}|\tilde{U})^{-1}P(\tilde{W}|x), \tag{3}$$

which means $pr(y|x)$ varies linearly with $P(\tilde{W}|x)$ under $\mathbb{H}_0$. In this regard, the linearity-based test in Miao et al. (2018) can be applied seamlessly to the continuous case.

However, the above approximations $P(y|\tilde{U}, x) \approx P(y|\tilde{U})$ and $P(\tilde{W}|\tilde{U}, x) \approx P(\tilde{W}|\tilde{U})$ may not hold, since the discretization of $U$ may break the independence structures. We call the $\|P(\tilde{W}|\tilde{U}, x) - P(\tilde{W}|\tilde{U})\|_1$ and the $\|P(y|\tilde{U}, x) - P(y|\tilde{U})\|_1$ as the discretization error, which can make the hypothesis testing invalid as the linearity no longer holds under $\mathbb{H}_0$.

In the subsequent section, we will characterize the smoothness Asmtion under which the discretization error can be reduced to an infinitesimal level.

## 4.2 CONTROLLING THE DISCRETIZATION ERROR

In this section, we provide the analysis regarding the discretization error control. Before preceding any technical detail, we first summarize our idea below. Specifically, the objective of interest is to find a proper way of discretization such that:

$$|pr(y|U \in B_j, x) - pr(y|U \in B_j)| \le \epsilon, \ \text{given } X \perp\!\!\!\perp Y|U,$$

for any $\epsilon > 0$. In this regard, applying to all $B_j \subset \mathrm{supp}(U)$, we have $|P(y|\tilde{U}, x) - P(y|\tilde{U})| \preceq [\epsilon]_{1 \times J}$ and $|P(\tilde{W}|\tilde{U}, x) - P(\tilde{W}|\tilde{U})| \preceq [\epsilon]_{K \times J}$, making the linearity in Eq. 3 approximately holds. For this purpose, we introduce the following smoothness conditions for functions $u \mapsto \mathcal{P}_{Y|U=u,X}$ and $u \mapsto \mathcal{P}_{W|U=u,X}$.

We assume the $u \mapsto \mathcal{P}_{Y|U=u,X}$, $u \mapsto \mathcal{P}_{Y|U=u}$, $u \mapsto \mathcal{P}_{W|U=u,X}$, and $u \mapsto \mathcal{P}_{W|U=u}$ are *resp.* $L_{Y|X}$-Lipschitz, $L_Y$-Lipschitz, $L_{W|X}$-Lipschitz, $L_W$-Lipschitz in terms of Total-Variation (TV) distance:

$$\mathrm{TV}(\mathcal{P}_{Y|U=u_1,X}, \mathcal{P}_{Y|U=u_2,X}) \le L_{Y|X}|u_1 - u_2|, \ \mathrm{TV}(\mathcal{P}_{Y|U=u_1}, \mathcal{P}_{Y|U=u_2}) \le L_Y|u_1 - u_2|,$$

$$\mathrm{TV}(\mathcal{P}_{W|U=u_1,X}, \mathcal{P}_{W|U=u_2,X}) \le L_{W|X}|u_1 - u_2|, \ \mathrm{TV}(\mathcal{P}_{W|U=u_1}, \mathcal{P}_{W|U=u_2}) \le L_W|u_1 - u_2|,$$

where $\mathrm{TV}(\mathcal{P}, \mathcal{Q}) := \frac{1}{2}\int |p(x) - q(x)|dx = \sup_D |\mathcal{P}(D) - \mathcal{Q}(D)|$.

*Remark* 4.2. This Asmtion was introduced in the conditional independence testing Warren (2021), however requires all variables to be observed. Essentially, this condition means the mapping $u \mapsto \mathcal{P}_{Y|U=u,X}, u \mapsto \mathcal{P}_{Y|U=u}, u \mapsto \mathcal{P}_{W|U=u,X}$, and $u \mapsto \mathcal{P}_{W|U=u}$ change smoothly with $u$.

---

[1]Note that $P(\tilde{W}|\tilde{U}, x) \approx P(\tilde{W}|\tilde{U}) \Leftrightarrow P(\tilde{W}|\tilde{U}, x)^{-1} \approx P(\tilde{W}|\tilde{U})^{-1}$, because the function $F(M) : M \mapsto M^{-1}$ from a matrix to its inversion is a continuous function.

To see how this Asmtion can control the discretization error of $\mathcal{P}_{Y|U,X}$ (the case for $\mathcal{P}_{W|U,X}$ is similar and we omit here for brevity), we note that under Asm. 4.2, it is easy to see that if the diameter of the bin $B_j$ is less than $\epsilon/L_Y$ (or $\epsilon/L_{Y|X}$), we will have $L_Y|u_1 - u_2| \leq \epsilon$ for any $u_1, u_2 \in B_j$. Then we have

$$\mathrm{TV}(\mathcal{P}_{Y|U=u_1}, \mathcal{P}_{Y|U\in B_j}) = \int_{B_j} \mathrm{TV}(\mathcal{P}_{Y|U=u_1}, \mathcal{P}_{Y|U=u_2}) d\mathcal{P}_U(u_2)/\mathcal{P}_U(B_j)$$
$$\leq \int_{B_j} L_Y|u_1 - u_2| d\mathcal{P}_U(u_2)/\mathcal{P}_U(B_j) \leq \epsilon.$$

According to the definition of TV-distance, we have

$$|pr(y|U = u_1) - pr(y|U \in B_j)| \leq \epsilon. \tag{4}$$

Similarly, if the diameter of $B_j$ is less than $\epsilon/L_{Y|X}$, we have

$$|pr(y|U = u_1, x) - pr(y|U \in B_j, x)| \leq \epsilon. \tag{5}$$

As we have $pr(y|U = u_1, x) = pr(y|U = u_1)$ given $X \perp\!\!\!\perp Y|U$, we combine (5) and (4) and use the triangle inequity to obtain that

$$|pr(y|U \in B_j, x) - pr(y|U \in B_j)| \leq 2\epsilon,$$

which means discretization error can be arbitrarily small when the bin diameter is small enough. We summarize the above analysis to the following theorem.

**Theorem 4.3** (Discretization error control with TV smoothness). *Suppose Asm. 4.2 holds. For each $\epsilon$, suppose that $\{B_j\}_{j=1}^{J}$ is a measurable partition of $\mathrm{supp}(U)$ such that for every bin $B_j$,*

$$diam(B_j) \leq \frac{1}{2}\min\{\epsilon/L_Y, \epsilon/L_{Y|X}\}.$$

*Then under $X \perp\!\!\!\perp Y|U$, we have:*

$$|pr(y|U \in B_j, x) - pr(y|U \in B_j)| \leq \epsilon.$$

*Remark* 4.4. Thm. 4.3 can be extended to accommodate weaker smoothness conditions such as the Hölder continuity. It can also be extended to cases where $U$ takes an unbounded support, by truncating the $U$ to a bounded support s.t. the probability of the residue term is smaller than than $\epsilon$.

Thm. 4.3 characterizes the condition for the discretization approach to be valid. In the following, we will show that such a smoothness condition can hold for the ANM model, which has been widely used in the causal inference literature Pearl (2009); Peters et al. (2017).

**Proposition 4.5** (ANM satisfying Lipschitz continuity). *Suppose that $X \leftarrow U \rightarrow Y$ (similarly for $X \rightarrow U \rightarrow Y$) is an addictive noise model, that is, $X = h(U) + N$ and $Y = g(U) + E$, where $N, E$ are exogenous noises. Denote the probability density function of $E$ as $f_E$. Then, the functions $u \mapsto \mathcal{P}_{Y|U=u,X}$ and $u \mapsto \mathcal{P}_{Y|U=u}$ are Lipschitz continuous with respect to $L_1$-distance if:*

1. *$g$ and $f_E$ are continuous and differentiable, and*

2. *$f_E'$ is absolute integral, i.e. $\int |f_E'(e)|de$ exists.*

*As for non-additive models, the 2nd condition can hold for the exponential family $E \sim exp$.*

In the following section, we will show how the discretization guarantee obtained above leads to a valid test of the null causal hypothesis.

### 4.3 HYPOTHESIS TEST WITH ASYMPTOTIC VALIDITY

In this section, we introduce our testing procedure for $\mathbb{H}_0 : X \perp\!\!\!\perp Y|U$ and show its asymptotic validity by leveraging results in Thm. 4.3. Specifically, given the smoothness condition outlined in Asmtion 4.2, when $\mathbb{H}_0$ holds, we can approximate $P(y|\tilde{U}, x)$ as $P(y|\tilde{U})$ and $P(\tilde{W}|\tilde{U}, x)$ as

$P(\tilde{W}|\tilde{U})$. This approximation allows us to assess the linear relationship $pr(y|x) \sim P(\tilde{W}|x)$ in (3), provided that the bin number is large enough to make each bin size sufficiently small.

To test the linearity, we discretize the proxy $W$ into $\tilde{W}$ and apply the procedure Miao et al. (2018) on $\tilde{W}$. Specifically, we denote $q_y := P(y|\tilde{X}), Q := P(\tilde{W}|\tilde{X})$ and $(\hat{q}_y, \hat{Q})$ be the corresponded estimators in Eq. 2. Similarly, we denote the least square residual of regressing $\hat{\Sigma}_y^{-\frac{1}{2}} \hat{q}_y$ on $\hat{\Sigma}_y^{-\frac{1}{2}} \hat{Q}$ as $\xi_y := \{I_n - \hat{\Sigma}_y^{-\frac{1}{2}} \hat{Q}^T (\hat{Q}\hat{\Sigma}_y^{-1}\hat{Q}^T)^{-1} \hat{Q}\hat{\Sigma}_y^{-\frac{1}{2}}\}\hat{\Sigma}_y^{-\frac{1}{2}} \hat{q}_y$. To ensure the uniqueness of the least square solution, we similarly assume Banerjee & Roy (2014); Miao et al. (2018): The matrix $Q$ has full row rank.

Then we construct the statistics $T_y := n^{-\frac{1}{2}} \xi_y^T \xi_y$ that converges to the chi-square distribution under $\mathbb{H}_0$, which provides the evidence to determine whether to reject the null hypothesis.

**Theorem 4.6** (Asymptotic validity). *Suppose models (a)-(b), Asm. 4.2, and Asms. 4.3 hold. Discretize $X, W$ to $I, K(I > K)$ levels with sufficient fine bins. Denote the level difference as $r := I - K$, the maximal bin diameter as $m^2$, and the sample size as $n$. If $\mathbb{H}_0 : X \perp\!\!\!\perp Y|U$ is correct, then $T_y \to \chi_r^2$ in distribution when $m \to 0, n \to \infty$.*

Thm. 4.6 can be generalized to account for all levels of $Y$. Specifically, suppose that $\tilde{Y}$ has $L$ levels and let $q^T := \{q_1^T, ..., q_{L-1}^T\}$. Then, under $\mathbb{H}_0$, we have:

$$q \approx \left\{ pr(y_1|\tilde{U})P(\tilde{W}|\tilde{U})^{-1}, ..., pr(y_{L-1}|\tilde{U})P(\tilde{W}|\tilde{U})^{-1} \right\} \begin{bmatrix} Q & 0 & 0 \\ \vdots & \ddots & \vdots \\ 0 & 0 & Q \end{bmatrix}.$$

Denote the diagonal matrix on the right-hand side as $Q_0$. $Q_0$ is a $I(L-1) \times I(L-1)$ matrix with full row rank. We can construct a new test statistic $T$ that aggregates all levels of $\tilde{Y}$ by replacing $(\hat{q}_y, \hat{Q})$ with $(\hat{q}, \hat{Q}_0)$ wherever they appear in the construction of $\xi_y$ and $T_y$. Specifically, suppose we have estimators $(\hat{q}, \hat{Q}_0)$ that satisfy:

$$n^{\frac{1}{2}}(\hat{q} - q) \to_d N(0, \Sigma), \ \hat{Q}_0 \to_p Q_0, \text{ and } \hat{\Sigma} \to_p \Sigma, \text{ with } \hat{\Sigma}, \Sigma \text{ positive-definite.} \quad (6)$$

Let $\xi := \{I - \hat{\Sigma}^{-\frac{1}{2}} \hat{Q}_0^T (\hat{Q}_0\hat{\Sigma}^{-1}\hat{Q}_0^T)^{-1} \hat{Q}_0\hat{\Sigma}^{-\frac{1}{2}}\}\hat{\Sigma}^{-\frac{1}{2}} \hat{q}$ be the least-square residual of regressing $\hat{\Sigma}^{-\frac{1}{2}} \hat{q}$ on $\hat{\Sigma}^{-\frac{1}{2}} \hat{Q}_0$ and $T := n\xi^T\xi$ be the test statistic, we have:

**Corollary 4.7.** *Assume models (a)-(b), Asm. 4.1, and Asms. 4.3. Discretize $X, W, Y$ to $I, K, L(I > K)$ levels. Denote $r := I(L-1) - K$, the maximal bin diameter as $m$, and the sample size as $n$. If $\mathbb{H}_0 : X \perp\!\!\!\perp Y|U$ is correct, then $T_y \to \chi_r^2$ in distribution when $m \to 0, n \to \infty$.*

*Remark* 4.8. The proposed test applies to any causal model that satisfies $X \perp\!\!\!\perp W|U$, including cases where $U, W$ are multivariate variables. It can also be generalized to accommodate observed confounders (mediators) $Z$, by re-defining $q_y := P(y|\tilde{X}, z)$ and $Q := P(\tilde{W}|\tilde{X}, z)$.

## 5 EXPERIMENT

In this section, we evaluate our method on synthetic data and a real-world application, *i.e.*, treatment study of sepsis disease in the Intensive Care Unit (ICU) scenario Johnson et al. (2016).

**Compared baselines.** We compare our method with the following baselines: **i) Vanilla** ($X \perp\!\!\!\perp Y|W$) that directly adjusts for the bias with the proxy variable $W$; **ii) Oracle** ($X \perp\!\!\!\perp Y|U$) which assumes the latent variable $U$ is available for conditioning; and **iii) Miao et al.** Miao et al. (2018) that was designed for discrete $X, U, W$ with respectively $I, J, K$ levels such that $I > J = K$.

**Metrics.** We report the type-I error and the type-II error of the test. The type-I error is the rate of rejecting a true null hypothesis, *i.e.*, the prediction is $X \to Y$ while the ground truth is $X \not\to Y$. The type-II error is the rate of not rejecting a false null hypothesis, *i.e.*, the prediction is $X \not\to Y$ while the ground truth is $X \to Y$.

**Implementation details.** The significant level $\alpha$ is set to 0.05. For our method, the $X, W, Y$ are discretized by quantile, with bin numbers (*resp.*) setting to $I = 16, K = 12, L = 5$ for synthetic data

---

[2] Here, $m$ is the maximal diameter of the (implicit) bins $\{B_j\}_{j=1}^J$.

and $I = 14, K = 12, L = 5$ for real-world data. The asymptotic estimator in Eq.2 is obtained by the empirical probability mass functions $\hat{pr}(w|x)$ and $\hat{pr}(y|x)$[3]. For the Vanilla and Oracle baselines, the kernel-based conditional independence (KCI) test Zhang et al. (2012) is used. For Miao et al. Miao et al. (2018), the discretization bin numbers are small and set to $I = 3, K = 2, L = 3$, so as to the necessity of sufficiently large bin numbers to control the discretization error.

## 5.1 SYNTHETIC DATA

**Data generation.** We consider the confounding graph and the mediation graph resp. shown in Fig. 2 (b) and (c) with $W \not\rightarrow Y$. For each graph, we use the structural equation $V_i = \sum_{j \in \mathbf{PA}_i} f_{ij}(V_j) + E_i$ to generate data for each vertex $V_i$ from its parent set $\mathrm{PA}_i$ and the associated exogenous noise $E_i$. For each $i$, we randomly choose $f_{ij}$ from $\{linear, tanh, sin, sigmoid\}$, and the distribution of $E_i$ from $\{gaussian, uniform, exponential, gamma\}$. To remove the effect of randomness, we repeat 10 times, with each time we generate 100 replications (*resp.*) under $\mathbb{H}_0$ and $\mathbb{H}_1$. The sample size is set to $\{400, 600, 800, 1000, 1200\}$.

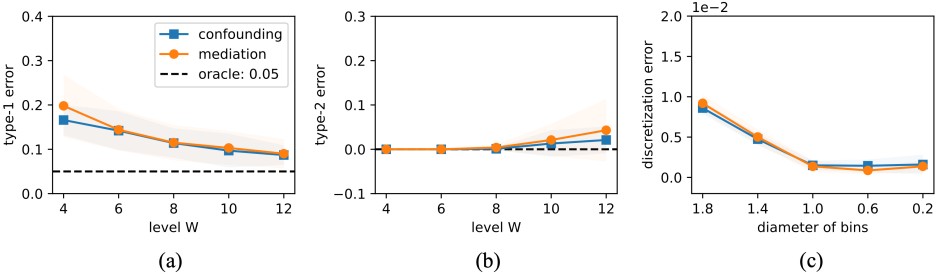

Figure 3: (a,b) Type-I and type-II errors of our method under different bin numbers. (c) Discretization errors under different bin diameters.

**Influence of bin numbers.** Fig. 3 shows the type-I and type-II errors of our method with bin numbers varying from 4 to 12. As we can see, the type-I error decreases and approaches the oracle level when $K$ increases. This is because when $K$ increases, the diameter of each bin becomes small. Under Asm. **??**, this means the discretization error $|pr(Y|U \in B_j) - pr(Y|U \in B_j, X)|$ is close to an infinitesimal level (as shown in Fig. 3 (c)) and $\hat{Q} \sim \hat{q}$ has an approximate linear relation. Therefore, the test statistic $T$ converges to the chi-square distribution, and the type-I error is controlled. For the type-II error as a trade-off with the type-I error, we can observe that it grows as the type-I error decreases. Besides, we can also observe that it is consistently small across different values of $K$, which may be due to the strong dependency between $X$ and $Y$.

**Hypothesis testing results.** Fig. 4 shows the type-I and type-II errors of our method and baselines. As shown, our method can control the type-I error to the threshold $p = 0.05$ up to a small inflation caused by the discretization error. Besides, the error gets smaller as the sample size grows, which aligns with our theoretical results in Thm. 4.6 and Cor. 4.7. Notably, the type-I error Miao et al. Miao et al. (2018) significantly bypasses the threshold as the bin numbers are too small to control the discretization error. It is also interesting to see from the result of the Vanilla method that directly conditioning on the proxy variable fails to remove the confounding bias caused by $U$.

## 5.2 TREATMENT OF SEPSIS DISEASE

We apply our method to infer the following causal relations in the scenario of sepsis disease:

1. Vancomycin $\rightarrow$ White Blood Cell (WBC) count. The vancomycin is a common antibiotic Moellering Jr (2006); Rybak et al. (2009) used to control bacterial infection (measured by the WBC count) in patients with sepsis. According to an existing RCT Rosini et al. (2015), vancomycin can causally influence the WBC count. As patients exhibit a higher mortality rate with an increased degree of bacterial infection, the white blood cell (WBC) count can serve as a surrogate measure of mortality, as illustrated in Figure 1 (a).

---

[3]Please refer to the appendix for details.

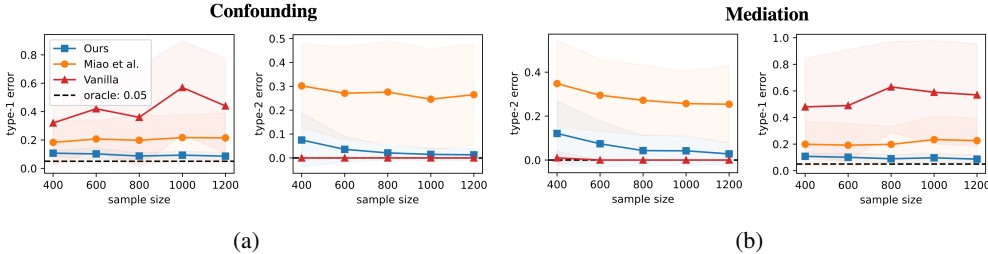

Figure 4: Type-I and type-II errors of our method and baselines under different sample sizes.

2. Morphine $\not\to$ WBC count. Prescription morphine is a common opioid medication used in pain management McQuay (1999). Clinically, the dosage of morphine appears a strong correlation with the WBC count Zhang et al. (2018). However, according to a recent RCT Anand et al. (2004), this correlation is highly likely to be confounding bias instead of causation.

For both cases, the latent confounder is the patient's health status. To adjust for the confounding, we use blood pressure as the proxy variable.

Table 1: Identified causal relations in sepsis disease.

| Method | Vancomycin$\to$ WBC | Morphine$\to$ WBC |
|---|---|---|
| Ground truth (RCT) | ✓ | ✗ |
| Vanilla | ✓ | ✓ |
| Miao et al.Miao et al. (2018) | ✗ | ✓ |
| Ours | ✓ | ✗ |

**Data extraction.** We consider the Medical Information Mart for Intensive Care (MIMIC III) Johnson et al. (2016) database, which consists of electronic health records from patients in the ICU. From MIMIC III, we extract 3,251 patients that are diagnosed with sepsis during their stays. Among these, there are (*resp.*) 1,888 and 559 patients with vancomycin and morphine records.

**Results.** Tab. 1 shows inferred causal relations of our method and baselines. As we can see, our results well align with the RCT findings in both cases, which demonstrates its effectiveness and utility. In contrast, there is a large discrepancy between the results given by baseline methods and the RCTs. For example, the vanilla method detects Morphine $\to$ WBC, which can be a false positive result due to the failure of adjusting for the confounding bias. Miao et al. Miao et al. (2018) gives inconsistent results in both cases, indicating that the proxy variable may lose its power without careful control of the discretization error.

## 6 CONCLUSION

In this paper, we propose a proxy-based method to identify causal relations when unobserved variables are present. Our method exploits the TV smoothness to show that the discretization error can be asymptotically controlled, rendering the hypothesis testing valid. We demonstrate the utility of our method on both synthetic data and the treatment study of sepsis disease.

**Limitation and future work.** Our method relies on discretization and therefore may suffer from accumulated discretization error for high-dimensional proxies since the error can be accumulated across dimensions. This is a common issue faced by all conditional independence (CI) test methods Ramdas et al. (2015); Shah & Peters (2020). To alleviate this problem, we will investigate theories on high-dimensional CI test Bellot & van der Schaar (2019); Ankan & Textor (2022) and discretization [Van Handel, 2014, Lem. 5.12] to pursue a delicate solution. Besides, our method can be potentially applied to multi-variable causal discovery methods by incorporating our method into existing algorithms such as the Fast Causal Inference (FCI) algorithm Spirtes et al. (2000).

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

APPENDIX

## A CONTROLLING THE DISCRETIZATION ERROR

In this section, we introduce the proof of Thm. 4.2 and Exam. 4.4.

For the convenience of the mathematical derivation, we use the $L_1$-distance and the Total Variance (TV) distance interchangeably, since for two distributions $\mathcal{P}, \mathcal{Q}$:

$$||\mathcal{P}, \mathcal{Q}||_1 := \frac{1}{2} \int |p(x) - q(x)| dx = TV(\mathcal{P}, \mathcal{Q}).$$

### A.1 PROOF OF THM. 4.2

**Theorem 4.2.** *Suppose that $u \mapsto \mathcal{P}_{Y|X,U=u}$ and $u \mapsto \mathcal{P}_{Y|U=u}$ are (resp.) $L_{Y|X}$-Lipschitz and $L_Y$-Lipschitz with respect to $L_1$-distance. Suppose that $\{B_j\}_{j=1}^J$ is a measurable partition of $supp(U)$, and that for every bin $B_j$,*

$$diam(B_j) \leq \frac{1}{2} \min\{\epsilon/L_Y, \epsilon/L_{Y|X}\}.$$

*Suppose that $X \perp\!\!\!\perp Y | U$, then, we have:*

$$|pr(y|U \in B_j, x) - pr(y|U \in B_j)| \leq \epsilon.$$

To prove Thm. 4.2, we first introduce the following lemma, which shows that if two distributions are close in $L_1$-distance, then their probability density values are also close.

**Lemma A.1.** *Consider two distributions $\mathcal{P}, \mathcal{Q}$ over the random variable $X$, if $||\mathcal{P}, \mathcal{Q}||_1 \leq \epsilon$, then $|pr_{\mathcal{P}}(X \in A) - pr_{\mathcal{Q}}(X \in A)| \leq \epsilon$, for any set $A \in supp(X)$.*

*Proof.* The proof is immediate considering the fact that the $L_1$-distance equals the TV distance and the definition that:

$$TV(\mathcal{P}, \mathcal{Q}) := \sup_A |pr_{\mathcal{P}}(X \in A) - pr_{\mathcal{Q}}(X \in A)|.$$

$\square$

We also require the following lemma, which connects the smoothness conditions to the closeness of distributions in $L_1$-distance.

**Lemma A.2.** *Suppose that $u \mapsto \mathcal{P}_{Y|U=u}$ be $L_Y$-Lipschitz with respect to $L_1$-distance. Suppose that $\{B_j\}_{j=1}^J$ is a measurable partition of $supp(U)$, and that $diam(B_j) \leq \epsilon/L_Y$ for every bin $B_j$ in the partition. Then, for all $u_0 \in B_j$, we have:*

$$||\mathcal{P}_{Y|U=u_0}, \mathcal{P}_{Y|U \in B_j}||_1 \leq \epsilon. \tag{7}$$

*Proof.* We first show that $\mathcal{P}_{Y|U \in B_j}$ can be written as:

$$\mathcal{P}_{Y|U \in B_j} = \fint_{B_j} \mathcal{P}_{Y|U=u} d\mathcal{P}_U(u), \tag{8}$$

where $\fint$ denotes the averaged integral over $B_j$, *i.e.*, the integral divided by the measure on the integration domain $B_j$.

Specifically, for any Borel set $D \subseteq supp(Y)$, the left hand side of (8) is:

$$\mathcal{P}_{Y|U \in B_j}(D) = pr(Y \in D | U \in B_j),$$

while the right-hand side is:

$$\int_{B_j} \mathcal{P}_{Y|U=u}(D) d\mathcal{P}_U(u) / \mathcal{P}_U(B_j) = pr(Y \in D, U \in B_j) / pr(U \in B_j).$$

Therefore, (8) holds according to Bayes' theorem.

Second, due to the assumed Lipschitzness, for any $u, u_0 \in B_j$, we have:

$$||\mathcal{P}_{Y|U=u_0}, \mathcal{P}_{Y|U=u}||_1 = \sup_D \left|\mathcal{P}_{Y|U=u_0}(D) - \mathcal{P}_{Y|U=u}(D)\right| \leq L_Y \left|u_0 - u\right| \leq \epsilon. \tag{9}$$

Now, to prove (7), we hope to prove:

$$||\mathcal{P}_{Y|U=u_0}, \mathcal{P}_{Y|U \in B_j})||_1 = \sup_D \left|\mathcal{P}_{Y|U=u_0}(D) - \mathcal{P}_{Y|U \in B_j}(D)\right| \leq \epsilon. \tag{10}$$

Next, we prove (10) by (8) and (9). Specifically, note that for any $D \subseteq supp(Y)$, we have:

$$
\begin{aligned}
|\mathcal{P}_{Y|U=u_0}(D) - \mathcal{P}_{Y|U \in B_j}(D)| &= \left|\mathcal{P}_{Y|U=u_0}(D) - \fint_{B_j} \mathcal{P}_{Y|U=u}(D) d\mathcal{P}_U(u)\right| && \text{\# (8)} \\
&= \left|\fint_{B_j} \mathcal{P}_{Y|U=u_0}(D) d\mathcal{L}_U(u) - \fint_{B_j} \mathcal{P}_{Y|U=u}(D) d\mathcal{P}_U(u)\right| && \text{\#} \fint_{B_j} d\mathcal{P}_U(u) = 1 \\
&\leq \fint_{B_j} \left|\mathcal{P}_{Y|U=u_0}(D) - \mathcal{P}_{Y|U=u}(D)\right| d\mathcal{P}_U(u) && \text{\# Triangle inequity} \\
&\leq \fint_{B_j} \epsilon \, d\mathcal{P}_U(u) = \epsilon. && \text{\# (9)}
\end{aligned}
$$

Since for any $D \subseteq supp(Y)$, $\left|\mathcal{P}_{Y|U=u_0}(D) - \mathcal{P}_{Y|U \in B_j}(D)\right| \leq \epsilon$, we have (10) holds, and hence prove the proposition. $\quad\square$

Analogously, we also have the following lemma.

**Lemma A.3.** *Suppose that $u \mapsto \mathcal{P}_{Y|X,U=u}$ be $L_{Y|X}$-Lipschitz with respect to $L_1$-distance. Suppose that $\{B_j\}_{i=1}^I$ is a measurable partition of $supp(U)$, and that $diam(B_j) \leq \epsilon/L_{Y|X}$ for every bin $B_j$ in the partition. Then, for all $u_0 \in B_j$, we have:*

$$||\mathcal{P}_{Y|X,U=u_0}, \mathcal{P}_{Y|X,U \in B_j}||_1 \leq \epsilon.$$

Equipped with Lem. A.1, Lem. A.2, and Lem. A.3, we now introduce the proof of Thm. 4.2.

*Proof of Thm. 4.2.* According to the Lipschitzness of $u \mapsto \mathcal{P}_{Y|X,U=u}$ and Lem. A.3, we have:

$$||\mathcal{P}_{Y|X,U=u_0}, \mathcal{P}_{Y|X,U \in B_j}|| \leq \frac{1}{2}\epsilon.$$

According to Lem. A.1, this indicates $|pr(y|x, U=u_0) - pr(y|x, U \in B_j)| \leq \frac{1}{2}\epsilon$.

Under $X \perp\!\!\!\perp Y | U$, we have $pr(y|x, U=u_0) = pr(y|U=u_0)$. Hence, we have:

$$|pr(y|U=u_0) - pr(y|x, U \in B_j)| \leq \frac{1}{2}\epsilon. \tag{11}$$

Similarly, according to the Lipschitzness of $u \mapsto \mathcal{P}_{Y|U=u}$, we have:

$$|pr(y|U=u_0) - pr(y|U \in B_j)| \leq \frac{1}{2}\epsilon. \tag{12}$$

Combining (11) and (12) with the triangle inequity, we have:

$$
\begin{aligned}
|pr(y|U \in B_j) - pr(y|x, U \in B_j)| &\leq \\
|pr(y|U \in B_j) - pr(y|U=u_0)| &+ |pr(y|U=u_0) - pr(y|x, U \in B_j)| \leq \epsilon.
\end{aligned}
$$

$\quad\square$

## A.2  PROOF OF PROP. A.4: ANMs THAT SATISFY THE LIPSCHITZ CONTINUITY

**Proposition A.4.** *Suppose that $X \leftarrow U \rightarrow Y$ (similarly for $X \rightarrow U \rightarrow Y$) is an addictive noise model, that is, $X = h(U) + N$ and $Y = g(U) + E$, where $N, E$ are exogenous noises. Denote the probability density function of $E$ as $f_E$. Then, the functions $u \mapsto \mathcal{P}_{Y|U=u,X}$ and $u \mapsto \mathcal{P}_{Y|U=u}$ are Lipschitz continuous with respect to $L_1$-distance if:*

*1. $g$ and $f_E$ are continuous and differentiable, and*

*2. $f'_E$ is absolute integral, i.e. $\int |f'_E(e)| de$ exists.*

*The second condition can be satisfied by, for example, the exponential family $E \sim exp$.*

*Proof.* Under $X \perp\!\!\!\perp Y|U$, we have $\mathcal{P}_{Y|X,U=u} = \mathcal{P}_{Y|U=u}$. Hence, it is suffice to show the Lipschitzness of $u \mapsto \mathcal{P}_{Y|U=u}$.

Specifically, we hope to prove $\forall u_1, u_2 \in supp(U)$:

$$||\mathcal{P}_{Y|U=u_1}, \mathcal{P}_{Y|U=u_2})||_1 \leq L_Y |u_1 - u_2|. \tag{13}$$

We first re-write (13) with the structural function $g$ and probability density function $f_E$.

Specifically, the $L_1$-distance can be re-written as the $L_1$-distance between the probability density functions:

$$||\mathcal{P}_{Y|U=u_1}, \mathcal{P}_{Y|U=u_2}||_1 = \frac{1}{2} \int \left| f_{Y|U=u_1}(y) - f_{Y|U=u_2}(y) \right| dy.$$

We also have:

$$f_{Y|U=u}(y) = f_E(y - g(u)),$$

by inversing the structural equation $Y = g(U) + E$.

As a result, (13) can be re-written as: $\forall u_1, u_2 \in supp(U)$,

$$\int |f_E(y - g(u_1)) - f_E(y - g(u_2))| \, dy \leq 2 L_y |u_1 - u_2|. \tag{14}$$

Next, we show (14) is true if:

1. $g$ and $f_E$ are continuous and differentiable,

2. $f'_E$ is absolute integrable, *i.e.* $\int f'_E(e) de$ exists.

Applying the Mean Value Theorem[4] on the function $u \mapsto f_E(y - g(u))$, we have $\exists u_c \in (u_1, u_2)$ s.t.

$$|f_E(y - g(u_1)) - f_E(y - g(u_2))| = |f'_E(y - g(u_c))g'(u_c)| \cdot |u_1 - u_2|$$

In this regard, we have:

$$\int |f_E(y - g(u_1)) - f_E(y - g(u_2))| \, dy = |g'(u_c)| \int |f'_E(y - g(u_c))| dy \, |u_1 - u_2|.$$

Let $L_Y := |g'(u_c)| \int |f'_E(y - g(u_c))| dy$[5], we have:

$$\int |f_E(y - g(u_1)) - f_E(y - g(u_2))| \, dy \leq 2 L_y |u_1 - u_2|,$$

hence, the Lipschitzness is proved.

$\square$

---

[4]The Mean Value Theorem requires the function $f_E \circ g$ to be continuous on $[u_1, u_2]$ and differentiable on $(u_1, u_2)$, which are satisfied by the assumed continuity and differentiability of $g$ and $f_E$.

[5]$L_Y$ is a constant value because we assume the function $f'_E$ is absolutely integrable.

# B  HYPOTHESIS TEST WITH ASYMPTOTIC VALIDITY

## B.1  PROOF OF THM. 4.8: ASYMPTOTIC VALIDITY

**Theorem 4.8.** *Assume models (a)-(b), Asm. 4.1, and Asms. 4.6-4.7. Discretize $X, W$ to $I, K(I > K)$ levels with sufficient fine bins. Denote the level difference as $r := I - K$, the maximal bin diameter as $m^6$, and the sample size as $n$. If $\mathbb{H}_0 : X \perp\!\!\!\perp Y | U$ is correct, then $T_y \to \chi_r^2$ in distribution when $m \to 0, n \to \infty$.*

*Proof.* According to the law of total probability, we first have:

$$pr(W \in C_k | x) = \sum_j pr(W \in C_k | U \in B_j, x) pr(U \in B_j | x).$$

In the form of the transition matrix, this means:

$$P(\tilde{W}|x) = P(\tilde{W}|\tilde{U}, x) P(\tilde{U}|x). \tag{15}$$

We also have:

$$pr(y|x) = \sum_j pr(y | U \in B_j, x) pr(U \in B_j | x).$$

In the form of the transition matrix, this means:

$$pr(y|x) = P(y|\tilde{U}, x) P(\tilde{U}|x). \tag{16}$$

Under Asm. 4.1, (15) can be rewritten as:

$$P(\tilde{W}|\tilde{U}, x)^{-1} P(\tilde{W}|x) = P(\tilde{U}|x). \tag{17}$$

(16) and (17) together give:

$$pr(y|x) = P(y|\tilde{U}, x) P(\tilde{W}|\tilde{U}, x)^{-1} P(\tilde{W}|x). \tag{18}$$

The idea is that, under Asm. 4.6 and $\mathbb{H}_0 : X \perp\!\!\!\perp Y | U$, we have $P(y|\tilde{U}, x) \approx P(y|\tilde{U})$ by Thm. 4.2.

In addition, under model (a)-(b) ($X \perp\!\!\!\perp W | U$), we also have $P(\tilde{W}|\tilde{U}, x) \approx P(\tilde{W}|\tilde{U})$.

Therefore, we have (informally):

$$pr(y|x) \approx P(y|\tilde{U}) P(\tilde{W}|\tilde{U})^{-1} P(\tilde{W}|x).$$

This means under $\mathbb{H}_0$, $pr(y|x)$ has a linear relationship with $P(\tilde{W}|x)$, for fixed $y$, as $x$ varies. Therefore, any evidence against the linearity is evidence against $\mathbb{H}_0$.

Formally speaking, rewrite (18) as:

$$
\begin{aligned}
&pr(y|x) \\
&= \left[ P(y|\tilde{U}) + P(y|\tilde{U}, x) - P(y|\tilde{U}) \right] \left[ P(\tilde{W}|\tilde{U})^{-1} + P(\tilde{W}|\tilde{U}, x)^{-1} - P(\tilde{W}|\tilde{U})^{-1} \right] P(\tilde{W}|x) \\
&= P(y|\tilde{U}) P(\tilde{W}|\tilde{U})^{-1} P(\tilde{W}|x) + \Delta P(\tilde{W}|x),
\end{aligned} \tag{19}
$$

where

$$
\begin{aligned}
\Delta = &\ \mathbb{P}(y|\tilde{U}) \left[ \mathbb{P}(\tilde{W}|\tilde{U}, x)^{-1} - \mathbb{P}(\tilde{W}|\tilde{U})^{-1} \right] + \left[ \mathbb{P}(y|\tilde{U}, x) - \mathbb{P}(y|\tilde{U}) \right] \mathbb{P}(\tilde{W}|\tilde{U})^{-1} \\
&+ \left[ \mathbb{P}(y|\tilde{U}, x) - \mathbb{P}(y|\tilde{U}) \right] \left[ \mathbb{P}(\tilde{W}|\tilde{U}, x)^{-1} - \mathbb{P}(\tilde{W}|\tilde{U})^{-1} \right].
\end{aligned}
$$

Since $X \perp\!\!\!\perp W | U$, Asm. 4.6 and Thm. 4.2, we have:

$$\lim_{\epsilon \to 0} P(\tilde{W}|\tilde{U}, x) - P(\tilde{W}|\tilde{U}) = [0],$$

---

[6] Here, $m$ is the maximal diameter of the (implicit) bins $\{B_j\}_{j=1}^J$.

where $[0]$ denote the zero matrix. Since matrix inversion $F(M) : M \mapsto M^{-1}$ is a continuous function, we have:

$$\lim_{\epsilon \to 0} P(\tilde{W}|\tilde{U}, x)^{-1} - P(\tilde{W}|\tilde{U})^{-1} = [0], \tag{20}$$

Under $\mathbb{H}_0 : X \perp\!\!\!\perp Y|U$, we also have:

$$\lim_{\epsilon \to 0} P(y|\tilde{U}, x) - P(y|\tilde{U}) = [0]. \tag{21}$$

With (20) and (21), we have:

$$\lim_{\epsilon \to 0} \Delta = [0],$$
$$\lim_{\epsilon \to 0} P(y|x) = P(y|\tilde{U})P(\tilde{W}|\tilde{U})^{-1}P(\tilde{W}|x).$$

Considering all bins of $supp(X)$, this can be written in the form of the transition matrix:

$$\lim_{\epsilon \to 0} q_y^T = P(y|\tilde{U})P(\tilde{W}|\tilde{U})^{-1}Q, \tag{22}$$

which means $q_y^T \sim Q$ (as $x$ varies) is linear under $\mathbb{H}_0$.

In the following, we discuss the asymptotic distribution of $n^{\frac{1}{2}}\xi_y$ and $T_y$.

We first show that $n^{\frac{1}{2}}\xi_y \to N(0, \Omega_y)$ in distribution when $n \to \infty$ and $\epsilon \to 0$. Specifically, given that $\hat{Q} \to Q$, $\hat{\Sigma}_y \to \Sigma_y$ in probability and $n^{\frac{1}{2}}(\hat{q}_y - q_y) \to N(0, \Sigma_y)$ in distribution, applying Slutsky's theorem, we have $n^{\frac{1}{2}}(\xi_y - \Omega_y\Sigma_y^{-\frac{1}{2}}q_y) \to N(0, \Omega_y)$ in distribution with $\Omega_y = I - \Sigma_y^{-\frac{1}{2}}Q^T(Q\Sigma_y^{-1}Q^T)^{-1}Q\Sigma_y^{-\frac{1}{2}}$. If $\mathbb{H}_0$ is correct, by (22), we have $\Omega_y\Sigma_y^{-\frac{1}{2}}q_y \to 0$, which, by applying Slutsky's theorem again, implies that $n^{\frac{1}{2}}\xi_y \to N(0, \Omega_y)$ in distribution.

Next, we show that $\Omega_y$ has rank $r = I - K$. Specifically, the rank of $Q$ is $K$ by Asm. 4.7. Because multiplying an invertible matrix does not change the rank of the matrix, the rank of $Q\Sigma_y^{-\frac{1}{2}}$ is $K$. Since transposing a matrix does not change its rank, the rank of the idempotent matrix $\Sigma_y^{-\frac{1}{2}}Q^T(Q\Sigma_y^{-1}Q^T)^{-1}Q\Sigma_y^{-\frac{1}{2}}$ is also $K$. By Corollary 11.5 of (Banerjee & Roy, 2014), the matrix $\Sigma_y^{-\frac{1}{2}}Q^T(Q\Sigma_y^{-1}Q^T)^{-1}Q\Sigma_y^{-\frac{1}{2}}$ has $K$ eigenvalues equal to one and $I - K$ eigenvalues equal to zero. Hence, $\Omega_y$ is an idempotent matrix with rank $r = I - K$.

Finally, we show that the test statistic $T_y := n^{\frac{1}{2}}\xi_y^T\xi_y$ has an asymptotic chi-square distribution with freedom $r$.

We first introduce a theorem about the asymptotic distribution of a random variable $X$ and the random variable $X^TX$. According to Thm. 1.12 of Shao (2003), if $X \to Y$ in distribution, then $nX^TX \to Y^TY$ in distribution. Now, let $X$ be $n^{\frac{1}{2}}$ and $Y$ be $N(0, \Omega_y)$, we have:

$$T_y := n\xi_y^T\xi_y \to N(0, \Omega_y)^T N(0, \Omega_y) \text{ in distribution.}$$

We then explain why $N(0, \Omega_y)^T N(0, \Omega_y) = \chi_r^2$. Because $\Omega_y$ is a symmetric matrix, we can find an orthogonal matrix $V$ such that $V\Omega_yV^T = diag(1, ..., 1, 0, ..., 0)$, a diagonal matrix with $r$ 1s and $I - r$ 0s in the diagonal. Hence, we have $VN(0, \Omega_y) \sim N(0, diag(1, ..., 1, 0, ..., 0))$ and,

$$N(0, \Omega_y)^T N(0, \Omega_y) = [VN(0, \Omega_y)]^T [VN(0, \Omega_y)] = \chi_r^2.$$

$\square$

## C  EMPIRICAL ESTIMATION METHOD

In this section, we provide the empirical estimation method to obtain estimators that satisfy the asymptotic properties:

$$n^{\frac{1}{2}}(\hat{q}_y - q_y) \to N(0, \Sigma_y) \text{ in distribution,}$$

$$\hat{Q} \to Q \text{ and } \hat{\Sigma}_y \to \Sigma_y \text{ in probability, with } \hat{\Sigma}_y, \Sigma_y \text{ positive-definite.}$$

We start from the univariate case for illustration.

**Univariate case.**  Consider a discrete random variable $X$ that takes $k$ levels of values $\{x_1, x_2, ..., x_k\}$. For $n$ iid samples $\{X_1, X_2, ..., X_n\}$, let $Z_i$ be the number of samples that take the value $x_i$. Then, we have $\mathbf{Z} := \{Z_1, Z_2, ..., Z_k\}^T$ follows a multinomial distribution with parameters $(n, \mathbf{p})$, where $\mathbf{p} := \{pr(x_1), pr(x_2), ..., pr(x_k)\}^T$. In the following, we use $p_i$ to denote $pr(x_i)$ for simplicity.

Our objective is to compute the Maximum Likelihood Estimator (MLE) $\hat{\mathbf{p}} := \{\hat{p}_1, ..., \hat{p}_k\}^T$ of $\mathbf{p}$, which enjoys the asymptotic properties.

For this purpose, we first write the joint probability mass function for $n$ sample:

$$f_n(z_1, z_2, ..., z_k | n, \mathbf{p}) = n! \prod_{i=1}^{k} \frac{p_i^{z_i}}{z_i!}.$$

Then, the log-likelihood is:

$$h_n(p_1, p_2, ..., p_k, n | \mathbf{z}) := log(f(z_1, z_2, ..., z_k | n, \mathbf{p})) = log(n!) + \sum_{i=1}^{k} z_i log(p_i) - \sum_{i=1}^{k} z_i!.$$

To obtain the MLE estimator $\hat{\mathbf{p}}$, we maximize $h_n(\mathbf{p})$, with $p_1 + p_2 + ... + p_k = 1$ as the constraint. This can be converted to an unconstrained optimization problem by a Lagrange multiplier:

$$\arg \max_{p_1, p_2, ..., p_k} l_n(p_1, p_2, ..., p_k) := h_n(p_1, p_2, ..., p_k, n) + \lambda(1 - \sum_{i=1}^{k} p_i).$$

The derivative $\nabla l_n(\mathbf{p})$ is:

$$\nabla l_n(\mathbf{p}) = [\frac{\partial l_n}{\partial p_1}, ..., \frac{\partial l_n}{\partial p_k}]^T = [\frac{z_i}{p_i} - \lambda, ..., \frac{z_k}{p_k} - \lambda]^T.$$

Let $\nabla l_n(\mathbf{p}) = \mathbf{0}$ and considering $\sum_i^k p_i^* = 1$, we have $p_i^* = \frac{z_i}{n}, \lambda = n$.

Therefore, the MLE estimator is $\hat{\mathbf{p}} = \{\frac{Z_1}{n}, ..., \frac{Z_k}{n}\}$. The covariance matrix of $\hat{\mathbf{p}}$ can be computed by definition:

$$\text{Cov}[\hat{\mathbf{p}}] = \frac{1}{n} \begin{bmatrix} p_1(1-p_1) & -p_1 p_2 & ... & -p_1 p_k \\ -p_2 p_1 & p_2(1-p_2) & ... & -p_2 p_k \\ \vdots & \vdots & & \vdots \\ -p_k p_1 & -p_k p_2 & ... & p_k(1-p_k) \end{bmatrix}. \tag{23}$$

Another way to compute the covariance matrix is to use the Fisher information. Specifically, the Taylor expansion of $\nabla l_n(\mathbf{p})$ at $\mathbf{p}_0$ (the ground-truth parameter) is:

$$\nabla l_n(\mathbf{p}) = \nabla l_n(\mathbf{p}_0) + [\overset{2}{\nabla} l_n(\mathbf{p}_0)]^T(\mathbf{p} - \mathbf{p}_0) + o(\mathbf{p}),$$

where $\nabla l_n(\mathbf{p}) = [\frac{z_1}{p_1} - n, ..., \frac{z_k}{p_k} - n]^T$, $\nabla^2 l_n(\mathbf{p}) = -\text{diag}[\frac{z_1}{p_1^2}, ..., \frac{z_k}{p_k^2}]$.

Let $\mathbf{p} = \hat{\mathbf{p}}$, considering the Hessian matrix $\nabla^2 l_n(\mathbf{p})$ is symmetric for any $\mathbf{p}$, we have:

$$\sqrt{n}(\hat{\mathbf{p}} - \mathbf{p}_0) = (-\frac{1}{n} \overset{2}{\nabla} l_n(\mathbf{p}_0))^{-1}(\frac{1}{\sqrt{n}} \nabla l_n(\mathbf{p}_0)).$$

Since the multinomial is defined on iid samples, and we have $l_n(\mathbf{p}) = \sum_i^n l_i(\mathbf{p})$, we can view $l_n(\mathbf{p})$ and its derivative as the summation of iid random vectors. Thus, by the law of large numbers and central limit (in the matrix form), we have (by noting that $\mathrm{E}[\nabla l_1(\mathbf{p}_0)] = 0$):

$$-\frac{1}{n}\overset{2}{\nabla} l_n(\mathbf{p}_0) \to -\mathrm{E}[\overset{2}{\nabla} l_1(\mathbf{p}_0)] \text{ in probability,}$$

$$\frac{1}{\sqrt{n}}\nabla l_n(\mathbf{p}_0) = \sqrt{n}(\frac{1}{n}\nabla l_n(\mathbf{p}_0) - \mathrm{E}[\nabla l_1(\mathbf{p}_0)]) \to N(\mathbf{0}, \mathrm{Var}[\nabla l_1(\mathbf{p}_0)]) \text{ in distribution.}$$

Let $J_n(\mathbf{p}) := -\mathrm{E}[\nabla^2 l_n(\mathbf{p})]$, $V_n(\mathbf{p}) := \mathrm{Var}[\nabla l_n(\mathbf{p})]$, we then have:

$$\sqrt{n}(\hat{\mathbf{p}} - \mathbf{p}_0) \to N(\mathbf{0}, J_1(\mathbf{p}_0)^{-1} V_1(\mathbf{p}_0) J_1(\mathbf{p}_0)^{-1}) \text{ in distribution,}$$

where $J_1(\mathbf{p}_0) = -\frac{1}{n}\mathrm{E}[\nabla^2 l_n(\mathbf{p}_0)] = -\mathrm{diag}\{\frac{1}{p_1}, ..., \frac{1}{p_k}\}$, and

$$V_1(\mathbf{p}_0) = \frac{1}{n}\mathrm{Var}[\nabla l_n(\mathbf{p}_0)] = \begin{bmatrix} \frac{1-p_1}{p_1} & -1 & ... & -1 \\ -1 & \frac{1-p_2}{p_2} & ... & -1 \\ \vdots & \vdots & & \vdots \\ -1 & -1 & ... & \frac{1-p_k}{p_k} \end{bmatrix}.$$

Note that the asymptotic covariance $J_1(\mathbf{p}_0)^{-1} V_1(\mathbf{p}_0) J_1(\mathbf{p}_0)^{-1}$ is consistent with the result in (23). Also note that $J_1$ no longer equals to $V_1$, that is, the Fisher information matrix has a "sandwich" form.

**Bivariate case.** Consider two discrete random variables $X, Y$, which *(resp.)* take $k, l$ levels of values $\{x_1, ..., x_k\}$ and $\{y_1, ..., y_l\}$. For $n$ iid samples $\{(X_1, Y_1), ..., (X_n, Y_n)\}$, let $Z_{ij}$ be the number of samples that take the value $(x_i, y_j)$. Then, we have $\mathbf{Z} := \{Z_{ij}\}_{ij}^T$ follows the multinomial distribution with parameters $(n, \mathbf{p})$, where $\mathbf{p} := \{pr(x_i, y_j)\}_{ij} = \{pr(x_i)pr(y_j|x_i)\}_{ij}$.

To obtain the MLE estimator $\hat{\mathbf{p}} = [\{\hat{p}(x_i)\}_i, \{\hat{p}(y_j|x_i)\}_{ij}]^T$, we again write the log-likelihood function for $n$ samples:

$$h_n(\mathbf{p}) = log(n!) + \sum_{i,j} z_{ij}log(p(y_j|x_i) + \sum_{i,j} z_{ij}log(p(x_i)) - \sum_{i,j} log(z_{ij}!). \qquad (24)$$

To obtain $\hat{\mathbf{p}}$, we maximize $h_n(\mathbf{p})$, with the following $k + 1$ constraints: $\sum_j p(y_j|x_i) = 1$ for $i = 1, 2, ..., k$, and $\sum_i p(x_i) = 1$. This can be converted to the following unconstrained optimization problem:

$$\arg\max_{\mathbf{p}} l_n(\mathbf{p}) := h_n(\mathbf{p}) + \lambda_1(1 - \sum_j p(y_j|x_1)) + ... + \lambda_k(1 - \sum_j p(y_j|x_k)) + \lambda_{k+1}(1 - \sum_i p(x_i)).$$

Let the derivative $\nabla l_n(\mathbf{p}) = [\{\frac{Z_{i:}}{p(x_i)} - \lambda_{k+1}\}_i, \{\frac{Z_{ij}}{p(y_j|x_i)} - \lambda_i\}_{ij}]^T = \mathbf{0}$, where $Z_{i:} := \sum_j Z_{ij}$, we have $\lambda_i = Z_{i:}, \lambda_{k+1} = n$, and the MLE estimator $\hat{\mathbf{p}} = [\{\hat{p}(x_i)\}_i, \{\hat{p}(y_j|x_i)\}_{ij}]^T$, where $\hat{p}(x_i) = \frac{Z_{i:}}{n}, \hat{p}(y_j|x_i) = \frac{Z_{ij}}{Z_{i:}}$.

In the following, we compute the covariance matrix of $\hat{\mathbf{p}}$ with the Fisher information. Specifically, we need to compute the matrices $J_1(\mathbf{p}_0)$ and $V_1(\mathbf{p}_0)$.

The first and second order derivative of $l_n(\mathbf{p})$ are *(resp.)*:

$$\nabla l_n(\mathbf{p}) = [\{\frac{Z_{i:}}{p(x_i)} - n\}_i, \{\frac{Z_{ij}}{p(y_j|x_i)} - Z_{i:}\}_{ij}]^T_{(k+kl)\times 1}$$

$$\overset{2}{\nabla} l_n(\mathbf{p}) = \mathrm{diag}[\{\frac{Z_{i:}}{p(x_i)^2}\}_i, \{-\frac{Z_{ij}}{p(y_j|x_i)^2}\}_{ij}]_{(k+kl)\times(k+kl)}.$$

Hence, $J_n(\mathbf{p}) = -\mathrm{E}[\nabla^2 l_n(\mathbf{p})] = n\mathrm{diag}[\{\frac{1}{p(x_i)}\}_i, \{\frac{p(x_i)}{p(y_j|x_i)}\}_{ij}]$.

By definition, we have $V_n(\mathbf{p}) = \mathrm{Var}[\nabla l_n(\mathbf{p})]$, which involves the following terms (obtained by simple calculus and the fact that $\mathrm{Cov}(A + B, C) = \mathrm{Cov}(A, C) + \mathrm{Cov}(B, C)$):

$$\mathrm{Var}[\frac{Z_{i:}}{p(x_i)} - n] = n \frac{1 - p(x_i)}{p(x_i)}$$

$$\mathrm{Var}[\frac{Z_{ij}}{p(y_j|x_i)} - Z_{i:}] = n \frac{p(x_i)(1 - p(y_j|x_i))}{p(y_j|x_i)}$$

$$\mathrm{Cov}[\frac{Z_{i1:}}{p(x_{i1})} - n, \frac{Z_{i2:}}{p(x_{i2})} - n] = -n$$

$$\mathrm{Cov}[\frac{Z_{i1j1}}{p(y_j1|x_i1)} - Z_{i1:}, \frac{Z_{i2j2}}{p(y_j2|x_i2)} - Z_{i2:}] = 0$$

$$\mathrm{Cov}[\frac{Z_{i1:}}{p(x_{i1})} - n, \frac{Z_{i2j}}{p(y_j|x_i2)} - Z_{i2:}] = 0.$$

Therefore, for $\hat{\mathbf{p}} = \{\hat{p}(x_1), ..., \hat{p}(x_k), \hat{p}(y_1|x_1), ..., \hat{p}(y_l|x_1), ..., \hat{p}(y_1|x_k), \hat{p}(y_l|x_k)\}^T$, we have:

$$\sqrt{n}(\hat{\mathbf{p}} - \mathbf{p}_0) \to_D N(\mathbf{0}, J_1(\mathbf{p}_0)^{-1} V_1(\mathbf{p}_0) J_1(\mathbf{p}_0)^{-1}),$$

where:

$$J_1(\mathbf{p}_0)^{-1} = \mathrm{diag}[\underbrace{p(x_1), p(x_2), ..., p(x_k)}_{k}, \underbrace{\frac{p(y_1|x_1)}{p(x_1)}, ..., \frac{p(y_l|x_1)}{p(x_1)}, ..., \frac{p(y_1|x_k)}{p(x_k)}, ..., \frac{p(y_l|x_k)}{p(x_k)}}_{k \times l}],$$

and

$$V_1(\mathbf{p}_0) = \begin{bmatrix} A & [0]_{k \times kl} \\ [0]_{kl \times k} & \mathrm{diag}\{a_{11}, a_{12}, ..., a_{kl}\}_{kl \times kl} \end{bmatrix},$$

with

$$A = \begin{bmatrix} \frac{1-p(x_1)}{p(x_1)} & -1 & \cdots & -1 \\ -1 & \frac{1-p(x_2)}{p(x_2)} & \cdots & -1 \\ \vdots & \vdots & & \vdots \\ -1 & -1 & \cdots & \frac{1-p(x_k)}{p(x_k)} \end{bmatrix}_{k \times k}.$$

Therefore, $J_1(\mathbf{p}_0)^{-1} V_1(\mathbf{p}_0) J_1(\mathbf{p}_0)^{-1}$ is:

$$\begin{bmatrix} B & [0]_{k \times kl} \\ [0]_{kl \times k} & \mathrm{diag}\{a_{11}b_{11}^2, a_{12}b_{12}^2, ..., a_{kl}b_{kl}^2\}_{kl \times kl} \end{bmatrix},$$

with

$$B = \begin{bmatrix} p(x_1)(1 - p(x_1)) & -p(x_1)p(x_2) & \cdots & -p(x_1)p(x_k) \\ -p(x_2)p(x_1) & p(x_2)(1 - p(x_2)) & \cdots & -p(x_2)p(x_k) \\ \vdots & \vdots & & \vdots \\ -p(x_k)p(x_1) & -p(x_k)p(x_1) & \cdots & p(x_k)(1 - p(x_k)) \end{bmatrix}_{k \times k}.$$

and

$$a_{ij} = \frac{p(x_i)(1 - p(y_j|x_i))}{p(y_j|x_i)}, b_{ij} = \frac{p(y_j|x_i)}{p(x_i)}.$$

In addition, according to the marginal property of the multinomial distribution, the asymptotic normality covariance $\Sigma_y$ of the estimator $\hat{q}_{y_j} = [\hat{p}(y_j|x_1), ..., \hat{p}(y_j|x_k)]^T = [\frac{Z_{1j}}{Z_{1:}}, ..., \frac{Z_{kj}}{Z_{k:}}]^T$ is:

$$\Sigma_y = \mathrm{diag}\{a_{1j}b_{1j}^2, ..., a_{kj}b_{kj}^2\} = \mathrm{diag}[\frac{p(y_j|x_1)(1 - p(y_j|x_1))}{p(x_1)}, ..., \frac{p(y_j|x_k)(1 - p(y_j|x_k))}{p(x_k)}],$$

whose consistent estimator $\hat{\Sigma}_y$ can be obtained by replacing $p$ with $\hat{p}$.

