# OpenReview forum: "Causal Discovery with Unobserved Variables: A Proxy Variable Approach"
_ICLR.cc/2024/Conference — ICLR 2024 Conference Withdrawn Submission_

### Official Review · Reviewer_4NAf · 2023-10-31

**Soundness:** 2 fair
**Presentation:** 2 fair
**Contribution:** 2 fair
**Rating:** 3
**Confidence:** 5

**Summary:**

This work proposes a hypothesis to identify the causal direction under the existence of unobserved variable. By assuming a proxy variable of the unobserved variables exists, these work try to extend the results in the discrete data (Miao et al. (2018)) to the continuous data by using the discretization.

**Strengths:**

- This work propose a proxy variable approach for identifying the causal relationship under the existence of unobserved variables.

**Weaknesses:**

- The contribution of this work seems somewhat limited as it only an extension of the previous work Miao et al. (2018) by using the discretization.
- This work supposes that the matrix P(W|U,x) is invertible after the discretization. However, unlike the discrete case, such condition can be possibly violate when the original data is continuous after being discretized, and it is necessary to discuss that in what condition and in which type of relationship that such invertbility holds.
- Moreover, based on the work in Miao et al. (2018), it seems that several additional assumptions are also required other than the invertable matrix one, and it is not disclosed and discussed in this work.
- In fact, instead of discretizing the data, is it possible to directly test the independence using the continuous information?

**Questions:**

See the weaknesses above.

---

### Official Review · Reviewer_iFkV · 2023-11-01

**Soundness:** 3 good
**Presentation:** 3 good
**Contribution:** 3 good
**Rating:** 6
**Confidence:** 3

**Summary:**

In this manuscript, a novel proximal-based hypothesis testing method has been proposed, and it comes accompanied by provable consistency. Notably, the authors have identified certain smoothness conditions that are compatible with several causal models, notably including Additive Noise Models. Experiments have been performed using both synthetic data sets and real-world data to validate the proposed method.

**Strengths:**

1. The manuscript does an excellent job of articulating the motivation behind the proposed method.

2. The analysis provided for the discretization is not only easy-to-follow but also enlightening, offering potential insights for readers in the domain.

**Weaknesses:**

1. The authors themselves have acknowledged a potential avenue of exploration: it would indeed be intriguing to see how the proposed test integrates with existing constraint-based methods. While this is not currently addressed, it presents an interesting direction for future research.

2. A notable omission is the lack of experimental evaluation or in-depth theoretical discussion concerning the scalability of the proposed method. This oversight might result in some reservations for practitioners considering the implementation of the method in expansive real-world situations.

**Questions:**

Could the authors elaborate on the specific assumption referenced in Sec 5.1? The section mentions, ‘Under Asm. ??, this means …’ but it isn't clear what this refers to.

---

### Official Review · Reviewer_eeri · 2023-11-01

**Soundness:** 3 good
**Presentation:** 1 poor
**Contribution:** 2 fair
**Rating:** 3
**Confidence:** 4

**Summary:**

Causal discovery methods do not work when there is a hidden confounder between two variables being tested. However, sometimes proxy variables (children of hidden confounders) can provide information about the hidden confounders, which can then be used to correctly identify causal relationships between variables. Previous work has attempted this but only for discrete variables. The current work attempts to find assumptions such that continuous variables that can be properly discretised such that the proxy causal discovery of previous work can be applied.

**Strengths:**

- The paper tackles an important problem, that is causal discovery in the presence of hidden confounders.

**Weaknesses:**

- The presentation is not great. There are numerous references to assumptions and models that are not well defined. The example 1.1 is entirely unclear, the details of it can be guessed at after reading the paper, but this is not a good thing. Figure 1b) is very unclear.
- I'm a bit unsure about the differences between previous works. It seems like the analysis and testing procedure are very similar to previous works. More specifically, it seems like two different works have been combined without too much novelty (see questions below).

**Questions:**

- What is figure 1b actually showing? Its not obvious that it is showing what you are claiming it is showing.
- Figure 1c. what independence is being measured here?
- Asm 4.1 is referred to multiple times, but there is no Asm 4.1 in the paper.
- Its not clear to me and it isn't explained why the discretisation can break the required independence structure? It will be useful if some intuition or reasoning is provided.
- Corollary 4.7: Where are models (a)-(b) defined?
- Given that the number of bins controls the trade-off between type 1 and type 2 errors, is there a heuristic for choosing this when a user does not have access to the ground truth?
- What exactly is the difference in Section 4.2 between your work and Warren (2021)? If the variable is unobserved, why does the theory of the previous work not hold in this section?
- In Section 4.3, what is the difference between your work and Miao et el (2018)? Is it just that you are applying a discretising procedure first?

---

### Official Review · Reviewer_iE3i · 2023-11-03

**Soundness:** 3 good
**Presentation:** 3 good
**Contribution:** 2 fair
**Rating:** 6
**Confidence:** 3

**Summary:**

The authors propose a method to extend the discrete proxy-based causal discovery method to continuous cases.  Their method is based on a comprehensive analysis regarding discretization error. The authors claim that the discretization error can be reduced to an infinitesimal level, provided the proxy is discretized with sufficiently fine bins.

**Strengths:**

The authors present a theoretical analysis of discretization error.  They also give a profound theoretical study on the asymptotic validity of the method.

**Weaknesses:**

There are several issues the authors need to address.
1. The paper is just an extension of an existing method, and the contribution of the paper is limited and incremental.

2.  The experimental study is not sufficient to validate the effectiveness of the method.  The authors could provide additional experimental results on multiple real-world datasets to show the benefits of the approach.

**Questions:**

The authors could add more experimental studies on additional real-world datasets to strengthen the paper.